# Prediction Model of Concrete Initial Setting Time Based on Stepwise Regression Analysis

**DOI:** 10.3390/ma14123201

**Published:** 2021-06-10

**Authors:** Wei-Jia Liu, Xu-Jing Niu, Ning Yang, Yao-Shen Tan, Yu Qiao, Chun-Feng Liu, Kun Wu, Qing-Bin Li, Yu Hu

**Affiliations:** 1State Key Laboratory of Hydroscience and Engineering, Tsinghua University, Beijing 100084, China; liuweijia@tsinghua.edu.cn (W.-J.L.); wonderfulnxj@tsinghua.edu.cn (X.-J.N.); qingbinli@tsinghua.edu.cn (Q.-B.L.); 2China Three Gorges Projects Development Co., Ltd., Chengdu 610017, China; yang_ning1@ctg.com.cn (N.Y.); tan_yaosheng@ctg.com.cn (Y.-S.T.); qiao_yu@ctg.com.cn (Y.Q.); liu_chunfeng@ctg.com.cn (C.-F.L.); wu_kun1@ctg.com.cn (K.W.)

**Keywords:** low-heat cement concrete, environmental factors, initial setting time, maturity, equivalent age, stepwise regression analysis

## Abstract

Mass concrete is usually poured in layers. To ensure the interlayer bonding quality of concrete, the lower layer should be kept in a plastic state before the upper layer is added. Ultimately, it will lead to the prediction of concrete setting time as a critical task in concrete pouring. In this experiment, the setting time of concrete in laboratory and field environments was investigated. The equivalent age of concrete at the initial setting was also analyzed based on the maturity theory. Meanwhile, factors affecting the setting time in the field environment were studied by means of multiple stepwise regression analysis. Besides, the interlayer splitting tensile strength of concrete subjected to different temperatures and wind speeds was determined. The results of laboratory tests show that both setting time and interlayer splitting tensile strength of concrete decrease significantly with the increase of air temperature and wind speed. In addition, the equivalent age of concrete at initial setting remains the same when subjected to different temperatures, while it decreases obviously with the increase of wind speed. In the field environment, the equivalent age of concrete at initial setting is greatly different, which is related to the variability of relative humidity and wind speed. The average air temperature and maximum wind speed are the main factors affecting the initial setting time of concrete. Furthermore, a prediction model is established based on the stepwise regression analysis results, which can predict the actual setting state in real-time, and hence controlling the interlayer bonding quality of dam concrete.

## 1. Introduction

The Wudongde and Baihetan Hydropower Stations in China’s Jinsha River Basin are classified as 300-m-level ultra-high concrete double-curved arch dams. These dams are currently the world’s largest, most complex hydropower projects under construction and face the most technical difficulties [1]. These key technical problems include temperature control, crack prevention and prevention of interlayer performance deterioration. These dams are geographically located in dry and hot valleys, which exhibit high temperatures, low humidity and strong winds. When mass concrete is poured in layers in a harsh environment, the interlayer bonding quality is often affected [2]. It is found that under the conditions of high wind and dry heat, concrete displays the phenomenon of less bleeding water and rapid evaporation, which eventually leads to the decrease of hydration degree, setting time, and vibrability of concrete [3,4]. In addition, the setting state of the lower concrete is considered closely related to the interlayer bonding strength [5]. Before the initial setting, the cement paste in concrete remains in a cohesive structure state, which exhibits thixotropic recovery. Suppose an upper concrete layer is poured before the initial setting of the lower concrete. In that case, the upper coarse aggregate sinks under gravity, and the coarse aggregates embed into each other at the interface between the upper and lower layers of concrete. Meanwhile, the cement paste at the layer surface is liquefaction vibrated, making the upper and lower layers of concrete into a unit. Consequently, its properties at the interlayer have little difference from that of inner concrete [6]. It is observed that when the interval time between layers is prolonged due to equipment failure and low efficiency of construction (i.e., mixing, pouring, paving, and vibrating), the differences in properties between the layer and inner concrete become more pronounced. Furthermore, in hot and dry valley areas, the harsh environment will accelerate the hardening process of concrete and significantly shorten the setting time, which leads to the deterioration of interlayer bonding performance [7]. Several studies about concrete interlayer properties have revealed that if the concrete surface is exposed for a long time, the interlayer mechanical properties, impermeability, and microstructure of the layer deteriorate to varying degrees [8,9,10,11,12].

Many methods have been used to obtain the setting time of concrete, such as penetration resistance, ultrasonic wave, electricity, maturity, the heat of hydration, and nuclear magnetic resonance. Penetration resistance is the standard test method for setting time. However, the traditional penetration resistance test requires wet screening of concrete, and the coarse aggregates after wet screening cannot be reused. That is, the penetration resistance test consumes a lot of manpower and material resources, which affects the construction progress.

The electrical methods can be further subdivided into resistivity, conductivity, and electromotive force methods. Li [13] and Xiao [14] investigated the hydration process of fresh concrete by means of resistivity. The former supposes that the time corresponding to the minimum value point and a turning point on the resistivity development curve is linearly related to the initial and final setting time; while the latter considers that the first peak value of the resistivity differential curve is the turning point of cement paste hardening, which is related to the final setting time of cement paste. Some studies [15,16] proposed using a derivative of the conductivity-time curve to predict the setting time of concrete. They found that the initial setting time corresponds to the highest peak of the conductivity differential curve. Based on electrochemical principle, Fang et al. [17] proposed a relationship between the setting time and the electromotive force of cement paste and proved the reliability of the electromotive force method in measuring the setting time of cement. However, the electrochemical reaction of electrodes will affect the accuracy of test results, so the electrical method is rarely used in practical construction.

The change of ultrasonic wave velocity reflects the hardening process of concrete. A setting time calculation method based on ultrasonic wave velocity can be established and compared with the setting time obtained by penetration resistance method. For cement paste without gas or of low air content, the initial setting time corresponds to the time when the P-wave velocity begins to increase [18]. The same conclusions have been reported by Gregor [19] and Carette [20]. However, the complexity of ultrasonic instrument and interference of clutter in the construction site can affects the accuracy of the test results. It is difficult to obtain satisfactory results in practical engineering application.

Based on the factors that affect concrete setting time, some researchers have provided prediction equations of concrete setting time by using mathematical techniques to train existing data. Ahmadi et al. [21] proposed the functional dependencies between the initial setting time and additives, ambient temperature, relative humidity, and wind speed by adapting multiple regression analysis. Mourad et al. [22] predicted that the key factors affecting the initial setting time of concrete are the type and content of admixtures, air temperature during concrete mixing, average and maximum values of air temperature before initial setting, average and maximum values of air humidity before initial setting, and maximum temperature of concrete. Based on the multiple regression analysis method, a prediction model for the initial setting time of concrete mixture under field conditions was established. In addition, Hu et al. [23] provided a comprehensive research study on the influences of retarder dosage, specific surface area of dead burned MgO powder, temperature, and water solid ratio on the setting time of magnesium phosphate cement via Bayes network. Meanwhile, a prediction model for the observation of setting time for magnesium phosphate cement was proposed by means of the regression.

Moreover, the influence of ambient temperature on setting time of concrete by using maturity method was first proposed by Pinto et al. [24]. Meanwhile, a method for estimating apparent activation energy (Ea) at the early hydration stage was proposed and verified through experiments. The results showed that the Ea value at the early stage of cement hydration can be estimated by setting time, and the maturity method can be used to calculate the setting time of concrete at different ambient temperatures. Similar conclusions have been reported by Garcia et al. [25]. Lachemi et al. [26] evaluated and tested different maturity functions in order to predict the initial setting time, and also provided a comparative study among various methods. The results showed that Freiesleben Hansen and Pederson (FHP) and Carino and Tank (CT) functions were superior to other functions in predicting setting time. Based on Pinto’s research results, Han et al. [27] used the maturity method to predict the setting time of concrete containing a super retarding agent (SRA) at different ambient temperatures. The results show that the setting time estimation method proposed by Pinto is suitable for concrete containing SRA.

In summary, maturity theory and multiple stepwise regression analysis are more convenient to predict concrete setting time. However, some shortcomings still exist. For instance, the maturity method only considers the effect of temperature on setting time, but ignores the influences of relative humidity and wind speed. In the case of multiple stepwise regression analysis method, previous studies did not consider the maximum value of environmental factors, which may affect the accuracy of the prediction model.

Therefore, in order to accurately perceive setting state of concrete, the setting time of low-heat cement concrete under different temperatures (i.e., 10, 22, 26 and 40 °C) and wind speeds (i.e., 0, 5 and 13 m/s) was studied based on maturity theory. In addition, on the basis of considering the maximum temperature, relative humidity and wind speed, the setting time of concrete in the field environment was analyzed by multiple stepwise regression analysis method. Meanwhile, the interlayer splitting tensile strengths of concrete subjected to different temperatures (i.e., 22, 26 and 40 °C) and wind speeds (i.e., 0, 5 and 13 m/s) were tested. The interlayer bonding quality was controlled according to the relationship between the layer state and interlayer splitting tensile strength of concrete.

## 2. Materials and Methods

### 2.1. Raw Materials

The mix proportions of dam concrete are given in Table 1. The water-binder ratio was 0.5. One-graded concrete was used in the laboratory tests. Four-graded concrete was used in the field tests. Low heat Portland (abbr. LHP) cement that is conforming to the Chinese National Standard GB175-2007 [28] was employed in all mixtures. Fly ash (abbr. FA) was used to reduce the heat of hydration. Coarse aggregate was composed of small gravel (5–20 mm), medium gravel (20–40 mm), large gravel (40–80 mm) and extra-large gravel (80–120 mm) with apparent densities of 2780, 2790, 2785 and 2780 kg/m^3^, respectively. Artificial sand, with the fineness modulus of 2.76 and apparent density of 2780 kg/m^3^ was used as fine aggregate. Further, retarded type II superplasticizer (water-reducing rate was about 19%) and concrete high-efficiency air-entraining agent (air content is about 4.3%), which meet Chinese National Standards GB8076-2008 [29], were also used in this experiment. Properties and composition of cement are listed in Table 2 and Table 3, respectively. The physical properties of fly ash are shown in Table 4. The concrete samples used for compressive strength test in Table 2 were cubes with the side length of 150 mm. Furthermore, stability refers to the uniformity of volume change after cement hardening. The stability test in the paper was carried out according to the Chinese standard GB/T1346-2011 [30] by using the Lei’s clip method.

### 2.2. Experimental Methods

#### 2.2.1. Penetration Resistance Tests in Laboratory

Penetration resistance tests of mortar were used accordingly with the China standard DL/T 5150-2017. The mortar was wet screened out of the concrete with a 5 mm sieve. Then pour mortar into cylindrical molds (ø100 mm × 150 mm). After vibrating and compacting, the samples were put into environmental simulation chambers (Figure 1a) with preset temperatures (10, 22, 26, and 40 °C) and wind speeds (0, 5 and 13 m/s). The wind speed was controlled by a turbocharged fan. During the process of penetration resistance test, specimens were not covered and bleeding was not removed (Figure 1b).

It should be noted that there is no difference between ASTM C403 [31] and Chinese standard DL/T 5150-2017 [32] in measuring concrete setting time by penetration resistance method. Since the design and construction of the Wudongde Dam adopts Chinese standards, the laboratory tests also need to be carried out in accordance with the requirements of Chinese standards.

This study comprehensively considers the actual environmental characteristics of the Wudongde dam project. Set the temperature to 0, 22, 26 and 40 °C according to the average temperature in different seasons. Set the wind speed to 5 m/s and 13 m/s according to the monthly average lower and higher wind speed.

According to the air temperatures (10, 22, 26, and 40 °C) and wind speeds (0, 5 and 13 m/s), specimens were classified into six groups: T10W0, T22W0, T26W0, T40W0, T22W5 and T22W13.

#### 2.2.2. Penetration Resistance Tests in Field Environment

Four-graded concrete was used in this project and lower gradation was adopted in some reinforcement areas. To achieve high accuracy, the setting time of four-graded concrete in the field environment was tested. The pouring area of each storehouse of the dam was kept up to 1100 m^2^, and the height of each storehouse was 3 m. The thickness of each layer of concrete was about 500 mm. After the concrete was poured, it needed to go through a paving and vibrating process, which lasted about 4 to 8 h. A small weather station in the field was used to collect the air temperature, relative humidity, wind speed, and rainfall information. The intelligent sprayers were installed around it to control the temperature and humidity of the storehouse, as shown in Figure 2a.

Concrete penetration resistance test was conducted on the storehouse surface. In the first stage, the four-graded concrete was wet screened, and then put into mortar barrels, as shown in Figure 2b. The penetration resistance test was conducted every half an hour.

The setting time of 9 storehouses of concrete was measured, and the initial temperature of concrete was controlled at about 8.9 °C (as shown in Table 6 presented later). It is noteworthy that the environmental temperature, humidity, and wind speed on the storehouse surface are not constant and change in real time, which are greatly different with those in the laboratory. According to the values of maximum air temperature, average temperature, maximum humidity, average humidity, maximum wind speed, and average wind speed before initial setting, a prediction model of initial setting time of concrete in the filed environment was established by means of stepwise regression analysis.

#### 2.2.3. Interlayer Splitting Tensile Strength Tests

Concrete was prepared according to Table 1 and poured in layers with mold size of 150 mm × 150 mm × 150 mm. The height of lower layer was 75 mm, as shown in Figure 3a. After pouring the lower layer, specimens were put into environmental simulation chambers with preset temperatures (22, 26, and 40 °C) and wind speeds (0, 5 and 13 m/s). The upper layer was poured after 6 h, as shown in Figure 3b. The interlayer splitting tensile strength of concrete was tested at the ages of 14 and 28 days. Each group was consist of three specimens, and the mean values were recorded.

The interlayer splitting tensile strength of concrete was conducted according to the Chinese standard DL/T 5150-2017 [32]. The specimens are cubes of 150 mm^3^. During the test, place specimens in the center of the lower plate of pressure testing machine. Then, a spacer between the upper and lower plates and specimens are placed. Afterwards, the specimens are split along the layer with a loading rate of 0.04 MPa/s~0.06 MPa/s (see Figure 3c).

According to air temperatures (10, 22, 26, and 40 °C) and wind speeds (0, 5, and 13 m/s), specimens were classified into five groups: C0, T22W0, T26W0, T22W5, and T22W13. Among them, group C0 refers to the concrete without layers, and the rest groups were poured in layers.

## 3. Results and Discussion

### 3.1. Results Obtained in Laboratory

#### 3.1.1. Mortar Penetration Resistance

Figure 4a shows the penetration resistance of mortar at different temperatures (i.e., 10, 22, 26 and 40 °C, wind speed 0 m/s). The penetration resistance of mortar changed slowly at low temperature (e.g., 10 °C), while it increased rapidly when the temperature raised. At 10, 22, 26 and 40 °C, the initial setting time of mortar were 13.5, 7.3, 6.0 and 3.1 h, while the final setting time were 25.9, 13.6, 11.1 and 5.7 h, respectively. It means that, the higher the air temperature, the faster the setting and hardening rate of mortar.

Figure 4b shows the penetration resistance of wet-screened mortar under different wind speeds (i.e., 0, 5 and 13 m/s). The penetration resistance of mortar increased slowly in windless conditions, while it increased rapidly with the increasing of wind speed. The setting time of mortar under different wind speeds were calculated, as shown in Table 4. At 0, 5, and 13 m/s, the initial setting time of mortar were 7.3, 4.4 and 4.2 h, while the final setting time were 13.6, 9.4 and 10 h, respectively.

In general, increasing the air temperature or wind speed can significantly accelerate the hardening process of mortar and shorten the setting time. It means that, both air temperature and wind speed are key factors affecting the setting time of mortar.

From the perspective of cement hydration, Freiesleben and Pedersen et al. [33] showed that the maturity of concrete is related to the degree of hydration and proposed a maturity Equation (2) based on the rate of hydration reaction based on the Arrhenius Equation (1).
(1)$$K_{T}=A\cdot \exp\left(-\frac{E_{a}}{\mathrm{RT}}\right)$$
where $K_{T}$ is the reaction rate constant; A is the frequency coefficient; R is the gas constant (8.314 J/K mol); T is the thermodynamic temperature (K, 273 + $T_{c})$, $T_{c}$ is the average temperature of concrete in a unit time interval; and $E_{a}$ is the apparent activation energy (J/mol). Further,
(2)$$t_{e}=\int_{0}^{t}\exp\left[\frac{E_{a}}{R}\left(\frac{1}{T_{r}}-\frac{1}{T}\right)\right]$$
where $t_{e}$ is the equivalent age, in hours or days; $T_{r}$ is the absolute temperature (293 K) at 20 °C.

Take the natural logarithm on both sides of Equation (1) results in Equation (3):(3)$$\ln K_{T}=\ln A-\frac{E_{a}}{R}\cdot \frac{1}{T}$$

According to Pinto et al. [24], the time required to reach the initial and final setting is inversely proportional to the hydration reaction rate constant, that is, $K_{T}\propto 1/t_{i}$, and $\ln K_{T}$ can be replaced by $\ln\left(1/t_{i}\right)$ in the Arrhenius equation:(4)$$\ln\left(1/t_{i}\right)=\ln A-\frac{E_{a}}{R}\cdot \frac{1}{T}$$
where $t_{i}$ is the initial setting time, and the temperature 22 °C, 26 °C, 40 °C and the apparent activation energy $E_{a}$ obtained according to Equation (4) can be brought into Equation (2) to obtain the corresponding equivalent age $t_{e}$ at the initial setting. For constant temperature conditions, Equation (5) can be applied to calculate the corresponding equivalent age at different temperatures under the time of initial setting:(5)$$t_{e}=t_{i}\cdot \exp\left[\frac{E_{a}}{R}\left(\frac{1}{T_{r}}-\frac{1}{T}\right)\right]$$

Similarly, the initial setting time at any constant temperature can be calculated through the equivalent age, as shown in Equation (6):(6)$$t_{i}{=t}_{e}\cdot \exp\left[-\frac{E_{a}}{R}\left(\frac{1}{T_{r}}-\frac{1}{T}\right)\right]$$

According to the values in Table 5 and the maturity function (Equation (5)), the equivalent age of concrete at initial setting can be calculated (reference temperature is 20 °C). The actual setting time and equivalent age at initial setting of concrete are shown in Figure 5a. With the increase of ambient temperature, the initial setting time of concrete gradually decreases, but the equivalent age of concrete at initial setting remains the same, which is basically consistent with the previous research [24,25]. However, the equivalent age of concrete at initial setting can be greatly shortened under windy conditions.

Replacing the average value of equivalent age in Table 4 into Equation (6), the initial time under any constant temperature can be seen in Figure 5b which shows the relationship between temperature and setting time of concrete. The results show that temperature is exponentially related to the setting time of concrete. For low-heat cement concrete in this research, the initial and final setting time can be calculated using Equations (7) and (8) without the influence of humidity and wind speed. The initial setting time is expressed by Time of Initial Setting (TIS), while the final setting time is expressed by Time of Final Setting (TFS):(7)$$\mathrm{TIS}=8.0\cdot \exp\left[-2855.1\times \left(\frac{1}{T_{r}}-\frac{1}{T}\right)\right]$$
(8)$$\mathrm{TFS}=15.1\cdot \exp\left[-2855.1\times \left(\frac{1}{T_{r}}-\frac{1}{T}\right)\right]$$

#### 3.1.2. Interlayer Splitting Tensile Strength

Based on the relationship between interval time and setting time, the layer state of concrete can be divided into three types: hot joint, warm joint, and cold joint, respectively. When the interval time between layers is less than the initial setting time, the layer is defined as the hot joint; the layer with the interlayer interval exceeding the initial setting time but without final setting is defined as the warm joint; the layer with the interlayer interval exceeding the final setting time is defined as the cold joint.

Figure 6 shows the variation of interlayer splitting tensile strength with different types of joint. The splitting tensile strength of layered concrete was lower than that of bulk concrete (C0). Compared with bulk concrete (C0) at 28 days, the interlayer splitting tensile strength of concrete with hot joint (T22W0) decreased by 20%, from 2.05 to 1.63 MPa. In the case of concrete with warm joint (T26W0, T22W5 and T22W13), the interlayer splitting tensile strength were 1.18, 0.96 and 0.64 MPa respectively, which was 42, 53, and 69% lower than that of C0. For the concrete with cold joint (T40W0), the strength (0.83 MPa) was only 40% of the bulk concrete (C0). In general, the interlayer performance of concrete deteriorated with the development of concrete layer state.

It should be noted that an abnormal phenomenon was observed in the experiment, as shown in Figure 6. The interlayer splitting tensile strength of concrete with cold joint (T40W0) was higher than that with warm joint (T22W13). This was mainly because the surface water content reduced to 80 and 100 kg/m^3^ respectively, when the mortar was subjected to 13 m/s and 40 °C for 6 h [7]. In other words, T22W13 lost more water than T40W0 in the same time. The reduction of water content leads to the decrease of hydration degree of concrete, which further affects the development of interlayer bonding strength. Therefore, it is not rigorous to evaluate the interlayer bonding quality of concrete only by the types of layer state under strong wind condition (i.e., 13 m/s). Further judgment should be made based on the water content of lower layer concrete.

The interlayer bonding quality will decrease seriously in the harsh construction environment. To ensure the interlayer splitting tensile strength above 80% of bulk concrete, it is necessary to control the layer state at the hot joint stage (that is, the interval time between layers is less than the initial setting time of concrete). Secondly, it is necessary to monitor the water content of the lower concrete to ensure that the concrete has enough water for hydration reaction.

### 3.2. Results Obtained in Field Environment

#### 3.2.1. Penetration Resistance

The equivalent age of concrete at initial setting under the storehouse surface environment was calculated based on Equation (2). The results are shown in Table 6.

The equivalent ages of concrete at the initial setting are different from that under constant temperatures. It is mainly attributed to the variability of relative humidity and wind speed under the storehouse surface environment, which affects both water evaporation and cement hydration process significantly. A previous research indicated that high wind speed can accelerate the dewatering rate of concrete. If the surface water content of concrete is less than total water requirement, the false setting will occur [34].

#### 3.2.2. Prediction of Initial Setting Time of Concrete in Field Environment

Due to the different equivalent ages of initial setting of concrete in field environment, maturity theory could not be applied to the complex storehouse surface environment. Therefore, from the data processing level, the functional relationship between the setting time of concrete and the environmental factors on the storehouse surface is established. It is assumed that the maximum temperature, average temperature, maximum relative humidity, average relative humidity, maximum wind speed, and average wind speed are the factors affecting the setting time of concrete before the initial setting. Taking the above six factors as independent variables and initial setting time as the dependent variable, the Statistical Product and Service Solutions (SPSS) software was used for stepwise regression analysis of the data series (see Table 7). SPSS is a professional data processing software, which can analyze and evaluate the data comprehensively. The main reason for adopting stepwise regression analysis is that it can screen out the variables that have a significant impact on the dependent variable (initial setting time) from the collinear independent variables. The condition for introducing the dependent variable is that the partial regression square sum of the factor is significant, which eliminates the dependent variables that have little influence on the independent variables from the model, to establish the optimal regression subset. This method not only solves the problem of collinearity, but also simplifies the regression equation.

Table 8 represents that the variance inflation factor (VIF) values of the average air temperature and the maximum wind speed before initial condensation are less than 5, indicating a lack of collinearity between the two independent variables.

T-test results proved that the associated probability of the regression coefficients of the maximum temperature, maximum relative humidity, average relative humidity and average wind speed before the initial setting of concrete was more significant than 0.05 (the standard value of the eliminated factors), which had been eliminated from the linear model. Therefore, the prediction model of the initial setting time of concrete under the storehouse surface environment could be expressed according to Equation (9):(9)$${\mathrm{TIS}=C}_{0}+\overset{2}{\underset{i=1}{\sum }}C_{i}{\times X}_{i}$$

In the field environment, the hydration process of concrete is mainly related to the ambient temperature and wind speed. On the one hand, the change of air temperature affects the hydration reaction rate of cementitious materials. On the other hand, higher temperature and wind speed accelerate the water evaporation rate of concrete, thus shortening the setting time of concrete.

By substituting the environmental parameters into Equation (9), the predicted values of setting time in the field environment can be obtained. Therefore, the setting time of concrete can be predicted according to the field environmental parameters

The methods mentioned in this study can be applied to the prediction of setting state of mass concrete in complex environments. However, the data used in the article is obtained from a specific environment, so the prediction model has certain limitations. In the future, more general models should be studied to predict the setting time of concrete in various environments.

## 4. Conclusions

To study the influence of environmental factors on the setting time of low-heat cement concrete, the penetration resistance tests in laboratory and field environment are carried out in this paper. The following conclusions are drawn:(1)Under the conditions of non-wind and constant humidity, the corresponding equivalent age of concrete in the initial setting in different temperature fields is the same. However, the equivalent age of concrete at the initial setting is different in the field environment which is related to humidity and wind speed.(2)In the field environment, the concrete setting state is influenced by the average air temperature and maximum wind speed before concrete initial setting. The prediction model established via stepwise regression analysis can accurately perceive the actual setting state of concrete.(3)The setting state of concrete and interlayer splitting tensile strength are significantly influenced by temperature. When pouring concrete at the construction site, the quality of interlayer bonding can be controlled based on the relationship between the concrete setting state and the splitting tensile strength.

## Figures and Tables

**Figure 1 materials-14-03201-f001:**
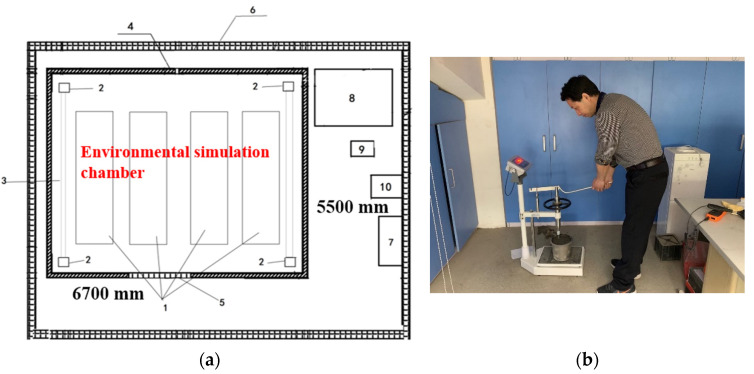
(**a**) Environmental simulation chamber. (**b**) Penetration resistance test. Note: 1—Test area, 2—embedded steel block, 3—hoisting system track, 4—polyurethane library board, 5—environmental simulation chamber entrance, 6—laboratory exterior wall, 7—computer, 8—compressor, 9—environmental simulation chamber control cabinet, 10—host control cabinet.

**Figure 2 materials-14-03201-f002:**
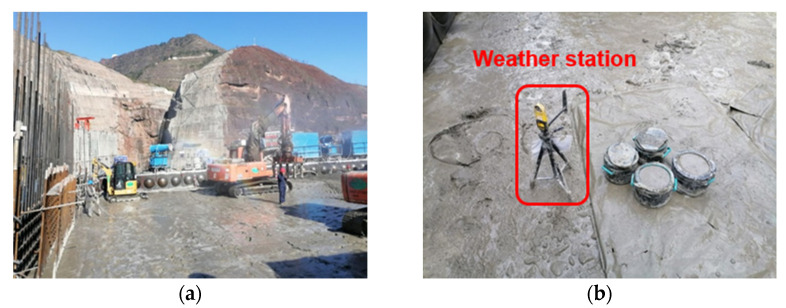
Construction site: (**a**) storehouse surface and spray equipment. (**b**) penetration resistance test and a small weather station.

**Figure 3 materials-14-03201-f003:**
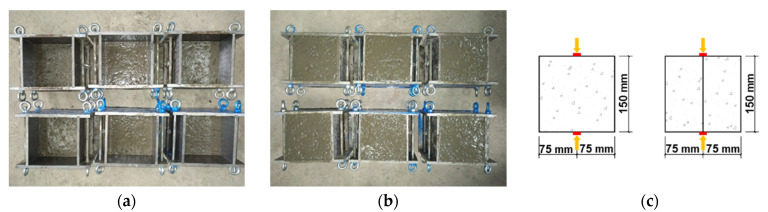
Concrete splitting tensile test: (**a**) Pouring lower layer concrete. (**b**) Pouring upper layer concrete. (**c**) Schematic diagram of splitting tensile test.

**Figure 4 materials-14-03201-f004:**
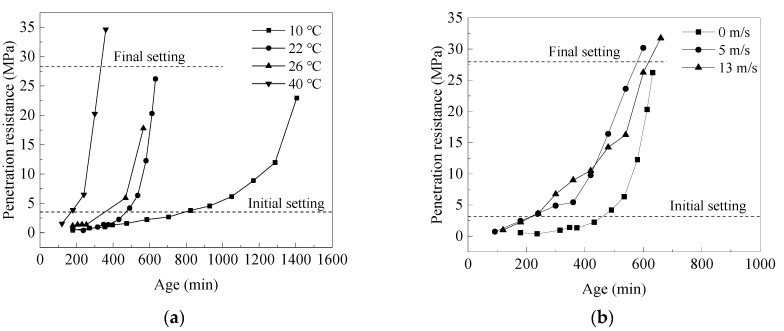
The change of penetration resistance of mortar under different environmental factors: (**a**) temperatures of 10, 22, 26 and 40 °C, non-windy, (**b**) 22 °C, wind speeds of 0, 5 and 13 m/s.

**Figure 5 materials-14-03201-f005:**
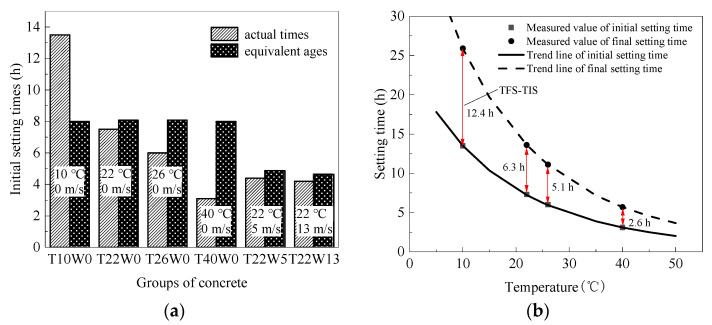
(**a**) Initial setting times and equivalent ages, (**b**) Variation curve of initial and final setting time of concrete at different temperatures.

**Figure 6 materials-14-03201-f006:**
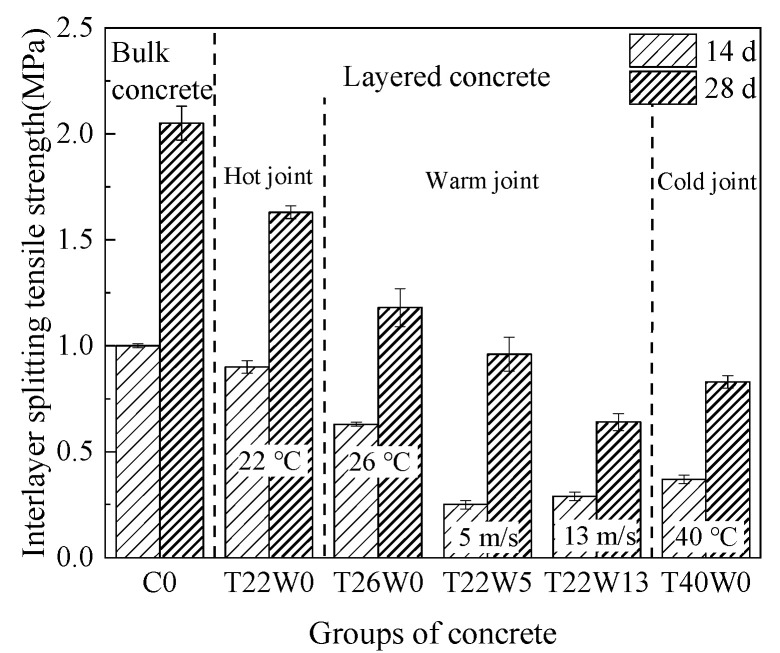
Comparison of interlayer splitting tensile strength in different layer states.

**Table 1 materials-14-03201-t001:** Mix proportion of concrete (kg/m^3^).

Concrete Gradation	Cement	Fly Ash	Water	Artificial Sand	Small Gravel	Medium Gravel	Large Gravel	Extra-Large Gravel	Superplast-Icizer	Air-Entraining Agent
One-graded	173	93	133	711	1320	-	-	-	1.596	0.0798
Four-graded	108	58	83	551	353	353	529	529	0.996	0.066

**Table 2 materials-14-03201-t002:** Physical properties of cement.

Density (g/m^3^)	Specific Surface (m^2^/kg)	Stability	Setting Time (min)		Compressive Strength (MPa)		
			Initial	Final	7 Days	28 Days	90 Days
3.23	324	qualified	216	279	20.9	47.2	69.8

**Table 3 materials-14-03201-t003:** Chemical compositions of LHP cement (% by mass).

CaO	SiO_2_	MgO	Al_2_O_3_	Fe_2_O_3_	SO_3_	Loss on Ignition	Alkali Content
61.64	23.24	4.74	4.07	3.02	2.05	1.08	0.45

**Table 4 materials-14-03201-t004:** Physical properties of fly ash.

Density (g/cm^3^)	Fineness/%	Water Content/%	SO_3_/%	f-CaO/%	Alkali Content/%	Loss on Ignition/%
2.35	8.2	0.2	0.57	0.15	1.26	3.79

**Table 5 materials-14-03201-t005:** The equivalent age of concrete at an initial setting under constant temperature.

Groups	Temperature (°C)	Relative Humidity (%)	Wind Speed (m/s)	TIS (h)	TFS (h)	Apparent Activation Energy (J/mol)	Equivalent Age at Initial Setting (h)	Average of Equivalent Age (h)
T10W0	10	30	0	13.5	25.9	36,118	7.9	8.0
T22W0	22	30	0	7.3	13.6		8.1	
T26W0	26	30	0	6.0	11.1		8.1	
T40W0	40	30	0	3.1	5.7		7.9	
T22W5	22	30	5	4.4	9.4	/	4.9	/
T22W13	22	30	13	4.2	10	/	4.6	/

**Table 6 materials-14-03201-t006:** Variation range of environmental variables on storehouse surface.

Number	Concrete Gradation	The Temperature of Concrete at the Mixer Outlet (°C)	Air Temperature Change Range (%)	Air Humidity Change Range (%)	Wind Speed Variation Range (m/s)	Initial Setting Time (h)	Equivalent Age at the Initial Setting of Concrete (h)
No. 1	Four-graded	8.4	25.8~30.3	54.0~90.0	0.0~4.4	6.4	7.5
No. 2	Four-graded	7.4	30.3~39.3	31.0~56.6	0.0~3.7	5.3	9.1
No. 3	Four-graded	9.7	23.5~37.3	26.7~85.8	0.0~5.3	6.4	9.2
No. 4	Four-graded	9.6	25.1~35.0	27.7~35.9	0.0~4.3	4.6	7.3
No. 5	Four-graded	9.4	26.2~29.0	38.7~44.0	0.2~6.3	5.4	6.2
No. 6	Four-graded	9.7	18.0~27.3	39.8~89.0	0.0~3.5	10.4	9.3
No. 7	Four-graded	8.4	28.0~32.8	56.2~78.5	0.0~2.9	6.8	9.1
No. 8	Four-graded	8.5	25.1~29.8	62.9~87.6	0.0~1.6	8.4	7.8
No. 9	Four-graded	8.8	21.2~33.4	41.0~93.0	0.0~2.1	9.3	9.5

**Table 7 materials-14-03201-t007:** Data samples of environmental variables on storehouse surface.

Number	Maximum Air Temperature (°C)	Average Air Temperature (°C)	Maximum Air Relative Humidity (%)	Average Relative Humidity of the Air (%)	Maximum Wind Speed (m/s)	Average Wind Speed (m/s)	Time of Initial Setting (h)
No. 1	30.3	28.3	90	63.6	4.4	1.1	6.4
No. 2	39.3	36.1	56.6	45.3	3.7	1.3	5.3
No. 3	37.3	30.0	85.8	47.7	5.3	1.8	6.4
No. 4	35.0	32.9	35.9	35.0	4.3	1.6	4.6
No. 5	29.0	27.6	44.0	41.9	6.3	2.3	5.4
No. 6	27.3	22.0	89.0	62.6	3.5	0.8	10.4
No. 7	32.8	30.6	78.5	62.0	2.9	0.68	6.8
No. 8	29.8	27.4	87.6	74.7	1.6	0.16	8.4
No. 9	33.4	25.5	93.0	52.3	2.1	0.41	9.3

**Table 8 materials-14-03201-t008:** Prediction equation of initial setting time of concrete under storehouse surface environment (R^2^ = 0.905).

i	Coefficient (C_i_)	Variable (X_i_)	Unit	VIF
0	19.649	-	(h)	-
1	−0.357	Average air temperature before initial setting	(°C)	1.034 (<5)
2	−0.616	Maximum wind speed before initial setting	(m/s)	1.034 (<5)

## Data Availability

The data presented in this study are available on request from the corresponding author.

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
