# Peer review of "Prediction Model of Concrete Initial Setting Time Based on Stepwise Regression Analysis"

_materials, 2021, doi:10.3390/ma14123201_

Round 1
Reviewer 1 Report
This article presents experimental test results on the setting times and an empirical equation for the estimation of the setting time. The topic and the results are of interest and importance in the concrete industry. This article needs to provide more details and some clarifications on their methods. Therefore, the reviewer recommends resubmitting the article with some revisions. The following is offered to assist the authors in their revisions:
- Briefly describe what the most technical difficulties are associated with the Wudongde dam (on pg. 1, ln 36).
- Can the authors describe a little more details of how they pour concrete in multi-layers in practice? The details can include the thickness of each layer, time, and volume.
- Provide the detail of the compositions of the Low heat Portland cement the authors used
- Is there any difference between ASTM C403 and China standard DL/T 5150-2017 for measuring the setting time of concrete by penetration method? Please explain why the authors conducted their test not according to the ASTM, if they are any reasons.
- Provide a picture of the Penetration resistance test setup including a picture of the environmental chamber.
- How did the authors control the wind speed?
- Why did the authors conduct the penetration resistance test in the lab on some samples only at 22 Celsius degrees with 5 and 13 m/s. Why not T40W5, T40W13 etc.
- Please provide some rationale why the authors designed their experiments at 0, 22, 26, 40 Celsius degrees and the wind speed of 0, 5, and 13 m/s.
- In Figure 1, please indicate the weather station the authors used in the field
- Is there a standard test method for interlayer splitting tensile strength?
- Please provide the description of the interlayer splitting tensile strength test the authors performed including how they tested the samples, what the loading rate was etc. A picture before and after the test would help readers understand what test the authors conducted.
- Please provide a little more insight into why the hydration process gets faster when the speed of wind is faster. The reviewer thinks when the wind blows faster, the circulation of the airflow gets faster, which would result in lowering the temperature.
- Why are the equivalent ages (h) at the initial setting time different between T22W0, T22W5, and T22W13? According to Eqs.1~5, the equivalent age at the initial setting time is not a function of ‘wind speed’.
- In Fig. 4b, the solid line and the dashed line is not calculation values. They are trend lines.
- As the authors stated on page 8, the water content is a crucial factor for the setting time. Please provide the w/cb ratio in Table 1.
- The sentence on pg 8, lns 286-287 is not quite clear to the reviewer. How did the authors reduce the surface water content to 80 and 100 kg/m3? According to Table 1, w/cb ratio was kept 0.5. Please clarify the situation.
- There are typos: e.g. m2 should be m^2 on pg.4, ln 171
Reviewer 2 Report
The manuscript presents physical testing (lab and field) and analysis to accurately perceive the setting state of concrete, the setting time of low-heat cement concrete under different temperatures and different wind speed.
The research can be interesting to the concrete communities including material scientists and civil engineers involved with the construction of concrete dams where the mass concrete is usually poured in layers.
The paper generally reads well and the method / results are clear but I have few comments:
For the data provided in Table 2, please clarify in the text the size of samples used for the compressive strength testing and elaborate further on the "stability" e.g. any particular type of test/ observation used to determine it was qualified.
The figures and data are clear and support the conclusion. Although the conclusion is generally ok but I would recommend the authors to expand on the impact of this study, its limitation and future direction.
Reviewer 3 Report
Congratulations for the findings from your research. Especially, Eq. 8 is reasonable and useful for the construction of the Chinese dam. In addition, the paper is easy to read and understand.
My comments are shown below. I am glad to help your revising the paper, even with my weak understanding.
- i) the results and conclusions from the field tests were independent from those of the laboratory tests. In other words, the highlight (Eq. 8) was obtained from a simple regression analysis, without data or the maturity theory described in Section 3.1.
ii) I could not find any new findings in the section of the laboratory tests (Section 3.1). The section is not necessary in the paper.
On the other hand, the highlight (Eq. 8) is useful for the actual construction of the Chinese dam. I think the application in the actual construction site is good.
---------------------
Introduction
It is recommended to add reviews on regression analysis with machine learning.
---------------------
page 4, line 180
"as shown in Table 5"
The layout of Table 5 is odd. The table is located far from page 4, line 180. I think the sentence should be changed to "as shown in Table 5 presented later".
---------------------
page 5, line 189
Could you add an explanation on the tensile strength testing. How did you introduce a tensile load to the specimens?
---------------------
page 9, line 323
What is "SPSS software"? Could you add a citation on SPSS software?
---------------------
page 10, Table 6
Table 6 is not cited in the paper.
-----------------------
Sections 3.1 (lab. tests) and 3.2 (field tests)
Could you explain the relationship between Sections 3.1 and 3.2? Was the equation 8 obtained without data of the laboratory tests or the maturity theory?
-----------------------
page 6, Figure 3.
Variations of three specimens should be shown and discussed.
-----------------------
pages 6-7, Eqs. (1)-(5), and Eqs (7a) and (7b)
Factors of wind speeds and humidity are not considered in the equations. What is the maturity theory for in the paper? At least, ideas/comments for improving the maturity theory are needed for the further study.
-----------------------
Sections 3.1 (lab. tests)
What are the new findings from Section 3.1?
-----------------------
page 9, Figure 5
Again, what are the new findings from Figure 5?
-----------------------
page 10, Eq. (8)
Could you add comments how to apply the evaluation equation for other construction sites? Is it only for the Chinese valley (at the Wudongde and Baihetan Hydropower Stations)?
Round 2
Reviewer 1 Report
The authors addressed most of the questions and suggestions. Since, the authors followed the Chinese standard methods for their testing, it is imperative to clarify why they used the Chinese standard in detail. The reviewer recommends adding a sentence stating that there is no difference between ASTM C403 and DT/L 5150-2017 in the manuscript as the authors responded in Question 4.
The reviewer did not find the figures of environmental chamber and the penetration resistance test setup provided in the response to Question 5. Please add them in the manuscript.
